# Effects of Foot Position-Based Gait Training on Muscle Activity, Gait Parameters, and Balance in Subacute Stroke Patients

**DOI:** 10.3390/healthcare12222206

**Published:** 2024-11-05

**Authors:** Yubin Lee, Yeongjae Pyo, Chaegil Lim

**Affiliations:** 1Department of Health Science, Gachon University Graduate School, Incheon 21936, Republic of Korea; edbql@naver.com; 2Myongji Chunhye Rehabilitation Hospital, Seoul 07378, Republic of Korea; dfyj8960@naver.com; 3Department of Physical Therapy, Gachon University, Incheon 21936, Republic of Korea

**Keywords:** curved gait, affected foot, muscle activity, gait parameters, balance ability

## Abstract

Background: the effects of gait training based on the positioning of affected foot muscle activity, gait parameters, and balance ability were investigated in patients with subacute stroke. Material and Methods: Forty-five patients with subacute stroke were randomly assigned to three groups: straight gait training (SGT) group (*n* = 15), outward curved gait training group (OCGT) with the paretic foot positioned laterally (*n* = 15), or inward curved gait training (ICGT) group with the paretic foot positioned medially (*n* = 15). All groups received 30 min interventions, comprising 15 min of gait training, five times per week for four weeks. Outcomes were measured in terms of muscle activation of the gluteus maximus (GM), vastus medialis, and vastus lateralis; five gait parameters (step length, stance phase, swing phase, velocity, and maximum force); and balance ability assessed using the timed up and go (TUG) test. Results: All groups exhibited significant improvements in all variables after the four-week intervention period (*p* < 0.05). Specifically, the overall muscle activation and gait parameters for each group increased as follows: the SGT showed increases of 38.8% and 5.7%, respectively; the OCGT exhibited improvements of 38.9% and 7.4%; and the ICGT demonstrated enhancements of 59.8% and 9.2%. However, except for comparisons between the SGT and ICGT groups in terms of GM muscle activity and TUG, no significant differences were observed between the groups for the other variables (*p* > 0.05). Conclusions: although patients with subacute stroke can improve their overall physical function regardless of the gait training method, ICGT may be more effective in enhancing muscle activity and balance ability.

## 1. Introduction

Stroke is one of the most common neurological disorders in the older population, and significantly impairs physical function and independence in daily life [1]. Stroke survivors often experience a range of neurological deficits, including alterations in consciousness, sensory and motor impairments, and altered muscle tone, all of which negatively affect their balance. Additionally, these patients may experience dysphagia and bowel dysfunction, further deteriorating their overall quality of life [2]. Among the various functional impairments, patients with stroke commonly exhibit increased spatiotemporal variability in gait, leading to a decline in lower limb function [3]. Consequently, these patients often present with a prolonged swing phase on the affected side, slower walking speed, and shorter stride length and width [4]. These patients also experience abnormal gait patterns and struggle with muscle activation and timing control, which further limits their functional walking ability [5].

In daily life, patients with stroke are required to perform straight gaits (SG) and curved gaits (CG), such as walking around furniture or navigating corners in hallways. CG accounts for approximately 30% of daily walking tasks and poses a more significant challenge for these patients [6,7].

CG requires precise coordination between body segments, including axial rotation control and maintenance of gaze in the direction of movement, which requires complex commands from the central nervous system [8,9]. Unlike SG, CG involves asymmetric lower-limb movements, with the outer leg covering a greater distance than the inner leg. Furthermore, maintaining propulsion during CG requires increased mediolateral ground reaction forces, and shifting body weight to the inner leg, and thereby requiring greater lower limb stability than during SG [10,11].

Chen et al. (2014) demonstrated that patients with stroke exhibit insufficient muscle activation and employ inefficient kinematic strategies during CG, leading to significantly reduced walking speeds compared to healthy individuals. Moreover, when navigating curves toward the affected side, these patients experience impaired function of the contralateral leg function, leading to a decrease in balance ability and an increased risk of falls [10]. Lee (2024) demonstrated that, compared to healthy controls, patients with stroke require phase-specific rehabilitation strategies during CG due to alterations in spatial synergy during the swing phase and temporal synergy during the swing–stance transition phase, depending on the walking direction [12]. The motor challenges encountered during CG in patients with stroke underscore the necessity for targeted training protocols tailored to CG patterns. Previous studies indicate that gait mechanics and functional adaptations vary significantly with foot placement and directionality in CG, highlighting the need to consider these variables in rehabilitation planning. However, recent studies primarily focused on analysis without empirical investigations assessing the clinical application of interventions in patient populations. The objective of this research is to evaluate the effects of training based on the positioning of the affected foot, comparing the impact on gait in both the CG and SG.

## 2. Materials and Methods

This study included 45 patients who were experiencing their first subacute stroke due to either cerebral infarction or hemorrhage and who were able to walk ≥6 m without assistive devices; however, individuals with visual impairments or vestibular disorders were excluded. This study was approved by the Institutional Review Board of Myongji Chun-Hye Rehabilitation Hospital (approval no. MJCHIRB-2023-05), and written informed consent was obtained from all participants before to the study.

Each participant was randomly assigned a number from 1 to 45 using a random function in Microsoft Excel for Windows (Microsoft 365, Version 12.0.6787.5000; Microsoft Corporation, Redmond, WA, USA) and then placed into one of three groups: the straight gait training (SGT) group (numbers 1–15, *n* = 15), the outside of the curved path gait training (OCGT) group (numbers 16–30, *n* = 15), and the inside of the curved path gait training (ICGT) group (numbers 31–45, *n* = 15). All groups underwent conservative physical therapy (CPT) and gait training for 30 min (15 min each) five times per week for four weeks (Figure 1). All interventions were supervised by therapists to prevent falls and accidents, and protective equipment (corner protectors, mattresses, etc.) was used to ensure safety. If the participants lost balance or reported dizziness during training, they were allowed to rest for one minute before resuming training.

### 2.1. Intervention

#### 2.1.1. CPT

CPT is based on neurodevelopmental therapy and includes joint mobilization exercises for the hip, knee, and ankle joints; lower limb muscle preparation exercises; and stability enhancement exercises through weight shifting prior to gait training. It also includes weight-shifting exercises in the standing position to improve the awareness of the paretic leg. Each session lasted five minutes.

#### 2.1.2. SGT

The SGT began with the participants seated, and they were trained to walk back and forth along a 20 m straight path. Participants maintained a forward gaze while walking, and after reaching the 20 m turning point, they turned in their preferred direction to repeat the 20 m walk [13]. All participants averaged 36.2 s in the pre-intervention timed up and go (TUG) test. To ensure participant safety and maintain continuous walking for 5 min, the time to complete the 20 m distance was maintained at ≤2 min.

#### 2.1.3. CG Training

CG training was conducted according to previous studies, where a circular path with ladder-shaped steps was used to adjust the stride length [13,14]. The walking path consisted of a large circle (total length: 18.8 m, radius: 3 m, and curvature: 0.75 m) and a small circle (total length: 12.6 m, radius: 2 m, and curvature: 0.5 m), both created using 1.7 cm wide tape. Additional radii were drawn 31 cm apart along the outer third of the path to create the ladder shape. A concentric inner circle was created by connecting points in the inner third of the radii, with the distance from the inner circle being 2 m for the large circle and 66 cm for the small circle (Figure 2).

Training began with the large circle in which participants walked along a curved path, placing their feet as close to the ladder-shaped markers as possible. If no balance issues or dizziness occurred during the 8 min session in the large circle, the training progressed to the small circle using the same method. To minimize the differences in walking distance, the time taken to complete one full rotation in the large circle was maintained at ≤1 min, and ≤30 s for the small circle. This procedure was repeated with foot placement adjusted according to the group assignment.

### 2.2. Assessment

After the four-week intervention, muscle activation of three muscles, five gait variables, and balance assessments were conducted. All assessments were performed by the same assessor before and after the intervention.

#### 2.2.1. Muscle Activity

Muscle activity was measured using a surface electromyography (EMG) system, Free EMG 1000 (BTS Bioengineering, Milan, Italy). Data were processed using an EMG Analyzer v2.9.37.0 (BTS Bioengineering, Milan, Italy). The EMG signals were sampled at a rate of 1024 Hz and filtered using a bandwidth of 20–500 Hz to remove the noise. Two adhesive electrodes (3M 2223H type) were used for each muscle. Prior to electrode attachment, the skin of the participant was shaved and cleansed with alcohol to minimize resistance and noise. Additional tape was used to secure the electrodes.

The measured muscles included three key components: the gluteus medius (GM), which provides stability to the lower limbs and pelvis during walking, maintains pelvic alignment, and generates mediolateral ground reaction forces to ensure balance of the center of mass; the vastus medialis (VM), which plays a significant role in push-off and affects knee extension and joint stability; and the vastus lateralis (VL), which is crucial during the initial phase of push-off, absorbing impact and controlling leg position during walking [15,16,17,18,19]. Electrode placement was performed following the SENIAM guidelines [20] (Table 1).

To standardize the data, EMG signals were measured for 5 s, and the raw data were converted to root mean square (RMS) values. Considering the participants were patients with stroke, the EMG signal values were expressed as a percentage of the voluntary reference contraction (%RVC). %RVC was calculated by dividing the RMS of the measured muscle (measurement value) by the RMS of the standard contraction and then converting it to a percentage. The RMS value for the standard contraction was measured and set using the average of 4 s, excluding 0.5 s before and after the 5 s period of standing still without movement. The RMS measurement value was obtained from the sit-to-stand movement without armrests, with the EMG signal being measured three times (1 min intervals) at the moment when the hips lifted off and touched the chair, and the average of these values was used.

#### 2.2.2. Gait Parameter

Gait parameters were measured using the Zebris FDM Treadmill System (Zebris Medical GmbH, Isny, Germany). Participants walked barefoot on the treadmill for 30 s, and five gait parameters were measured for the paretic leg using Zebris FDM-T software (version 1.18.44): step length, stance phase, swing phase, velocity, and maximum force (Table 2). All gait parameters, except for the swing phase, indicated an improvement in gait function when they increased. The Zebris FDM-T system has high reliability, with an intraclass correlation coefficient (ICC) of 0.97 [21].

#### 2.2.3. TUG Test

The TUG test assesses the dynamic balance, functional mobility, and gait ability in patients with stroke. The test includes sitting, standing, and turning, making it a quick assessment tool and predictor of gait function [22]. The participants were seated in an armchair, stood up at the signal of the evaluator, walked 3 m, turned around, and sat back down. The time required was recorded using a stopwatch [23]. The TUG test has high test–retest reliability (ICC = 0.95) [24]. The test was performed thrice, and the average value was used.

### 2.3. Statistical Analysis

All statistical analyses were performed using the SPSS software (version 23.0; IBM, Armonk, NY, USA). Data for all variables were expressed as mean ± standard deviation. Normality was assessed using the Shapiro–Wilk test. Paired *t*-tests were used for within-group comparisons and one-way ANOVA was used for between-group comparisons. The significance level was set at α = 0.05.

## 3. Results

No significant differences were observed in general characteristics between the groups (Table 3). The results of muscle activity, gait variables, and balance ability before and after the 4-week intervention are presented in Table 4 and Table 5. Significant changes were observed in the activity of all muscles across all groups (*p* < 0.00), with the GM showing a significant difference only between the SGT and ICGT groups. However, there were no significant differences in the VM and VL between the groups.

All groups showed significant differences in all gait parameters before and after the intervention; however, no significant differences were observed between the groups.

Regarding the balance assessment, there were significant differences between all groups before and after the intervention (*p* < 0.00), and a significant difference was observed only between the SGT and ICGT groups in the intergroup comparison (*p* < 0.00).

## 4. Discussion

The effects of a 4-week CG training based on the placement of the affected foot on lower limb muscle activity, gait variables, and balance ability in patients with subacute stroke were investigated in this study. Although significant changes were observed in all the groups before and after the intervention, some differences were observed depending on the training method and foot placement. The initial hypothesis was that the ICGT would demonstrate greater improvements in muscle activity, gait variables, and balance ability than the other groups; however, this was only partially supported by the results.

During CG, weight is transferred to the leg on the inner side of the curved path, requiring greater lower limb stability than during SGT [11]. Additionally, GM activation may improve body awareness by facilitating pelvic rotation and weight transfer to the inner body [14]. In this study, both the VL and VM significantly improved within the groups before and after the intervention, but no significant differences were observed between the groups. This indicates that VM and VL muscle activation can be enhanced regardless of the training method, and this increased muscle activity likely contributes to improved knee stability and ground reaction forces during gait [25,26]. The results demonstrate that the stance phase and ground reaction force increased by 4.5% in the SGT group, and by 5.3% and 6.7% in the OCGT and ICGT groups, respectively. This improvement was linked to an increase in step length, with the average step length increasing by approximately 7 cm and 11 cm in the SGT and CG training groups, respectively. This resulted in an overall post-intervention increase in gait speed of approximately 0.4 m/s across all groups [27,28]. Tilson et al. (2010) estimated 0.16 m/s improvement in gait speed as the minimal clinically important difference for subacute stroke interventions, which supports the clinical significance of these improvements in both SGT and CG training [29]. However, the SGT proved more effective in enhancing the muscles that contribute to knee stability, with VL and VM muscle activity increasing by 25.5% and 5.1%, respectively, indicating a particular advantage of SG in this area. Within the CG group, the OCGT showed greater improvements in VL and VM muscle activity, with increases of 2.6% and 23.6%, respectively, compared to the ICGT, suggesting that the OCGT places higher stability demands on the outer leg.

While GM activity also improved significantly within each group following intervention, a notable difference was found only between the SGT and ICGT groups. This suggests that the ICGT results in a greater increase in the weight-bearing capacity of the affected leg, leading to improved GM activation and potentially enhanced gait ability [17]. Furthermore, the improvement rate of VL and VM was higher in SGT, while GM activation was higher in CG, with ICGT showing a 10% greater increase than OCGT.

In summary, these results suggest that SG is optimal for enhancing knee stability, while CG training, particularly ICGT, provides greater benefits for improving weight-bearing capacity. Therefore, ICGT is recommended when the goal is to enhance weight transfer, load distribution, and sensory feedback, which may further contribute to improvements in balance and overall gait performance.

In the TUG test, which assessed balance ability, all groups demonstrated significant reductions in test times post-intervention, indicating improvements in dynamic balance. Pournajaf et al. (2023) and Lin et al. (2022) reported improvements in TUG ranging from 12.6 to 13.4 s following 15 to 20 sessions of robot-assisted gait training. Similarly, in this study, the ICGT exhibited a reduction of 12.5 s, reflecting effects comparable to robot-assisted training [30,31]. Furthermore, Arabzadeh et al. (2023) reported that increased GM activation leads to enhanced pelvic stability, which in turn improves balance ability by stabilizing the proximal lower limb [32]. In this study, GM activation increased by 18% in the SGT group and 50% in the ICGT group, with a 42% greater improvement in the ICGT group. Consequently, this enhanced the GM activation and positively influenced TUG performance, with the ICGT demonstrating an 18.7% greater improvement in balance ability compared to the SGT, thus highlighting a significant difference between the two groups.

Moreover, during walking, the eyes and head typically face the direction of travel, guide trunk movement, and facilitate weight transfer, leading to improvements in dynamic balance and gait ability. However, during CG, the gaze is directed toward the inner side of the curve as the person walks [8,9,33]. This suggests that, during CG, the trunk shifts toward the leg on the inside of the curved path, promoting weight transfer. Jin and Song (2017) revealed that when weight transfer to the inner leg increases, the load on that leg also increases, providing more afferent sensory information that can improve weight-shifting ability and dynamic balance [14]. This may explain the greater improvement in balance ability in the ICGT group than in the SGT group, as the ICGT group experienced increased weight transfer to the affected foot.

Overall, gait parameters showed only minor differences between groups, ranging from 3% to 11%, with gait speed differences of up to 0.2 m/s, suggesting that all gait training methods led to positive improvements for patients with subacute stroke. For targeted outcomes, the SGT is recommended for enhancing knee stability, while OCGT is advisable within CG training for similar goals. However, if the focus is on pelvic stability and overall weight-bearing capacity, the ICGT within the CG training is preferable. The results suggest that, since the aspects of improvement continue to vary depending on the gait training methods, no statistically significant differences are considered to exist between groups for most of the variables. In particular, the ICGT appears numerically superior in improving gait variables and balance capacity, suggesting enhanced efficacy in supporting weight transfer, load distribution, and sensory feedback.

This study had several limitations. First, the curvature was not varied at regular intervals during the intervention period. Second, the muscle activation was measured only in the gluteus medius, vastus medialis, and vastus lateralis. Future studies should measure the activation of other stability muscles, considering changes in curvature as well as movements of the trunk and upper limbs.

## 5. Conclusions

The effects of 4-week CG training based on the placement of the affected foot on lower limb muscle activity, gait variables, and balance ability in patients with subacute stroke were investigated in this study. The results suggest that muscle activity, balance ability, and gait performance can improve regardless of the gait training method. However, training with the affected foot positioned on the inner side of the curve was more effective in enhancing muscle activation and balance ability.

## Figures and Tables

**Figure 1 healthcare-12-02206-f001:**
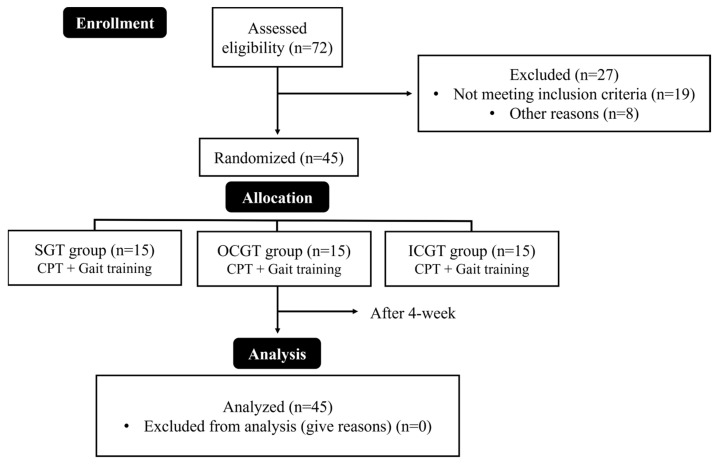
Flow chart.

**Figure 2 healthcare-12-02206-f002:**
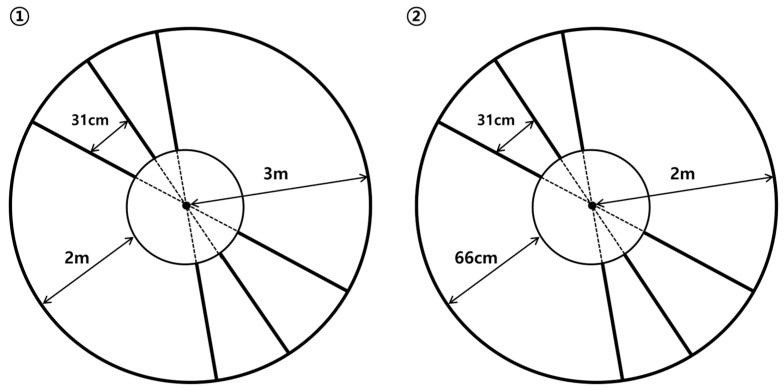
Curved walking path: ① large circle; ② small circle.

**Table 1 healthcare-12-02206-t001:** Muscles and attachment regions measured using sEMG.

Muscle	Attachment Region
GM	Located 1/2 the distance between the iliac crest and the greater trochanter of femur.
VM	Distal 80% on the line from the anterior superior iliac spine to the joint space in front of the anterior border of the medial collateral ligament.
VL	Proximal 1/3 on the line from the anterior superior iliac spine to the superior part of the patella.

GM, gluteus medius; VM, vastus medialis; and VL, vastus lateralis.

**Table 2 healthcare-12-02206-t002:** Gait variables measured using the Zebris gait analyzer.

Variable	Definition
Step length (cm)	Distance between the heels of both feet
Stance phase (%)	Proportion of contact with the ground
Swing phase (%)	Proportion of time the foot is in the air
Velocity (km/h)	Applied treadmill speed
Maximum force (N/kg)	Maximum pressure detected by the pressure sensor, normalized by the subject’s body weight

**Table 3 healthcare-12-02206-t003:** General characteristics of participants.

Category	STG	OCGT	ICGT	*p*
Age (years)	56.5 ± 13.9	56.5 ± 12.2	55.3 ± 15.7	0.959
Gender (male/female)	8/7	8/7	7/8	0.915
Height (cm)	165.5 ± 5.8	164.1 ± 8.6	166.1 ± 7.7	0.752
Weight (kg)	67.3 ± 11.8	65.3 ± 13.9	66.7 ± 11.7	0.934
Stroke type (infarction/hemorrhagic)	8/7	9/6	9/6	0.497
Paralyzed side (right/left)	8/7	6/9	9/6	0.537
Post-stroke duration (month)	4.1 ± 1.2	4.0 ± 0.8	4.1 ± 1.2	0.639

Data are expressed as mean ± standard deviation.

**Table 4 healthcare-12-02206-t004:** Comparison of muscle activity, gait variables, and balance within group on 4-week intervention.

Outcome Measurement	Variable	STG (*n* = 15)			OCGT (*n* = 15)			ICGT (*n* = 15)		
		Pre	Post	*p* *	Pre	Post	*p* *	pre	Post	*p* *
Muscle activity (%RVC)	GM	228.6 ± 194.9	281.2 ± 216.5	0.043	116.5 ± 52.4	194.0 ± 99.6	0.000	124.7 ± 43.4	249.6 ± 94.0	0.000
	VL	176.8 ± 86.8	271.4 ± 91.7	0.002	144.31 ± 49.1	183.1 ± 55.4	0.009	172.1 ± 80.9	214.0 ± 77.8	0.004
	VM	143.8 ± 44.7	209.9 ± 77.8	0.001	186.2 ± 92.6	294.5 ± 128.9	0.000	169.1 ± 85.7	227.6 ± 108.0	0.000
Gait variables	Step length (cm)	25.0 ± 6.1	31.7 ± 8.5	0.000	27.7 ± 5.9	38.3 ± 4.4	0.000	30.5 ± 9.9	41.4 ± 8.1	0.000
	Stance phase (%)	66.2 ± 4.0	70.7 ± 2.2	0.002	66.2 ± 3.8	71.5 ± 3.7	0.003	66.8 ± 3.4	73.5 ± 4.5	0.000
	Swing phase (%)	33.8 ± 4.0	29.3 ± 2.2	0.002	33.8 ± 3.8	27.3 ± 4.3	0.001	32.5 ± 4.1	26.4 ± 4.7	0.003
	Velocity (km/h)	0.6 ± 0.2	1.0 ± 0.3	0.000	0.8 ± 0.2	1.1 ± 0.2	0.000	0.7 ± 0.2	1.2 ± 0.4	0.000
	Maximum force (N/kg)	8.9 ± 0.8	9.4 ± 0.9	0.003	9.0 ± 1.0	9.5 ± 0.9	0.000	9.9 ± 1.1	10.8 ± 1.1	0.020
Balance	TUG	34.9 ± 8.7	29.1 ± 7.6	0.000	38.5 ± 11.8	29.9 ± 11.4	0.000	35.26 ± 12.6	22.8 ± 9.5	0.000

Data are expressed as mean ± standard deviation. STG, straight gait training; OCGT, outside curved gait training; ICGT, inside curved gait training; GM, gluteus medius; VL, vastus lateralis; VM, vastus medialis; and TUG, timed up and go. * *p* < 0.05.

**Table 5 healthcare-12-02206-t005:** Comparison of muscle activity, gait variables, and balance between groups on 4-week intervention.

Outcome Measurement	Variable	Significant Difference (*p*)		
		Between the STG and OCGT	Between the STG and ICGT	Between the OCGT and ICGT
Muscle activity (%RVC)	GM	0.396	0.027 *	0.080
	VL	0.055	0.066	0.851
	VM	0.109	0.704	0.291
Gait variables	Step length (cm)	0.109	0.152	0.955
	Stance phase (%)	0.669	0.242	0.515
	Swing phase (%)	0.290	0.430	0.872
	Velocity (km/h)	0.377	0.495	0.093
	Maximum force (N/kg)	0.842	0.979	0.847
Balance	TUG	0.070	0.003 *	0.098

STG, straight gait training; OCGT, outside curved gait training; ICGT, inside curved gait training; GM, gluteus medius; VL, vastus lateralis; VM, vastus medialis; and TUG, timed up and go. * *p* < 0.05.

## Data Availability

The data used in this study are available upon request from the corresponding author. The data are not publicly available due to privacy or ethical considerations.

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
