# Peer review of "Effects of Foot Position-Based Gait Training on Muscle Activity, Gait Parameters, and Balance in Subacute Stroke Patients"

_healthcare, 2024, doi:10.3390/healthcare12222206_

Round 1

Reviewer 1 Report

Comments and Suggestions for Authors

This study investigates the effects of foot-position-based gait training on muscle activity, gait parameters, and balance in subacute stroke patients. It compares three different training methods—straight gait, outward curved gait, and inward curved gait.

Abstract:
The abstract provides a concise summary of the study. However, it could benefit from including more specific statistical outcomes, such as the percentage of improvement in muscle activity and balance across the different training groups (Abstract, Page 1).

Methods:

  • Clarification Needed: The methodology is well-outlined, but it would be helpful to clarify whether the same assessors conducted pre- and post-intervention evaluations to ensure data consistency (Methods, Page 3, Section 2.2).
  • More information on the selection criteria for the participants, such as how randomization was performed and whether participants were matched for factors like age and physical ability, would improve transparency. This would possibly help readers better understand how participants were assigned to groups and ensure that the comparison between groups was balanced and fair. (Methods, Page 3, Section 2.1).

Results:

  • It would be useful to clarify why no significant differences were found between the groups for many parameters. Providing an explanation would assist in interpreting the results (Results, Page 6, Table 5).
  • Including the minimal clinically important difference (MCID) for parameters such as gait speed, step length, and stance phase would help readers understand the clinical significance of the changes (Results, Page 5, Table 2).
Discussion:
  • Highlighting on the implications for physical therapists and rehab professionals would be beneficial for these professionals to possibly implement results into practice.
Limitations:
  •  The study acknowledges some limitations, but further elaboration on how the exclusion of upper body movements may have impacted the results (Discussion, Page 9)
Minor Edits:
  • Table 1: Consider including confidence intervals for muscle activity and gait parameters to provide a clearer picture of the variability in the results (Results, Page 6, Table 1).

Author Response

Comments 1. Abstract: The abstract provides a concise summary of the study. However, it could benefit from including more specific statistical outcomes, such as the percentage of improvement in muscle activity and balance across the different training groups (Abstract, Page 1).

Response 1: Thank you for the detailed point-out. We have revised the content as follows:

In the original statement," All groups exhibited significant improvements in all variables after the 4-week intervention period (p < 0.05)." we have made the following modifications:

 "All groups exhibited significant improvements in all variables after the 4-week intervention period (p < 0.05). Specifically, the overall muscle activation and gait parameters for each group increased as follows: SGT group showed increases of 38.8% and 5.7%, respectively; the outside curved gait training OCGT group exhibited improvements of 38.9% and 7.4%; and the inside curved gait training ICGT group demonstrated enhancements of 59.8% and 9.2%." (Abstract, page 1, line 21-25)

Comments 2. Methods:

2-1. Clarification Needed: The methodology is well-outlined, but it would be helpful to clarify whether the same assessors conducted pre- and post-intervention evaluations to ensure data consistency (Methods, Page 3, Section 2.2).

2-2. More information on the selection criteria for the participants, such as how randomization was performed and whether participants were matched for factors like age and physical ability, would improve transparency. This would possibly help readers better understand how participants were assigned to groups and ensure that the comparison between groups was balanced and fair. (Methods, Page 3, Section 2.1).

Response 2: Thank you for the detailed point-out. We have revised the content as follows:

2-1. In Section 2, we added the following:

“After the four-week intervention, muscle activation of three muscles, five gait variables, and balance assessments were conducted. All assessments were performed by the same assessor before and after the intervention” (2.2 Assessment, page 4, line 140-142)

2-2. The participant selection criteria were modified from: "This study included 45 patients with subacute stroke and either cerebral infarction or hemorrhage who were able to walk ≥ 6 m without assistive devices" to: "This study included 45 patients who were experiencing their first subacute stroke due to either cerebral infarction or hemorrhage and who were able to walk ≥ 6 m without assistive devices; however, individuals with visual impairments or vestibular disorders were excluded" (Materials and Methods, page 2, line 74-77)

Additionally, we specified the random assignment of participants as follows:

 "Each participant was randomly assigned a number from 1 to 45 using a random function in Microsoft Excel for Windows (Microsoft Corporation, Redmond, WA, USA) and then placed into one of three groups: the straight gait training (SGT) group (numbers 1–15, n = 15), the outside of the curved path gait training (OCGT) group (numbers 16–30, n = 15), and the inside of the curved path gait training (ICGT) group (numbers 31–45, n = 15)." (Materials and Methods, page 2, line 79-84)

Comments 3. Results:

3-1.  It would be useful to clarify why no significant differences were found between the groups for many parameters. Providing an explanation would assist in interpreting the results (Results, Page 6, Table 5).

3-2. Including the minimal clinically important difference (MCID) for parameters such as gait speed, step length, and stance phase would help readers understand the clinical significance of the changes (Results, Page 5, Table 2).

Response 3: Thank you for the detailed point-out. We have revised the content as follows:

3-1. We revised the discussion section to explain the lack of significant differences among groups across various variables. This revision takes into account the overall flow of the text and additional considerations. We have added this explanation in three parts as follows:

  • "In summary, these results suggest that SG training is optimal for enhancing knee stability, while CG training—particularly ICGT—provides greater benefits for improving weight-bearing capacity. Therefore, ICGT is recommended when the goal is to enhance weight transfer, load distribution, and sensory feedback, which may further contribute to improvements in balance and overall gait performance." (Discussion, page 9, line 273-277)
  • "However, the SGT proved more effective in enhancing the muscles that contribute to knee stability, with VL and VM muscle activity increasing by 25.5% and 5.1%, respectively, indicating a particular advantage of SG in this area. Within the CG group, the OCGT showed greater improvements in VL and VM muscle activity, with increases of 2.6% and 23.6%, respectively, compared to the ICGT, suggesting that the OCGT places higher stability demands on the outer leg" (Discussion, page 9, line 261-266)
  • "Overall, gait parameters showed only minor differences between groups, ranging from 3% to 11%, with gait speed differences of up to 0.2 m/s, suggesting that all gait training methods led to positive improvements for patients with subacute stroke. For targeted outcomes, the SGT is recommended for enhancing knee stability, while OCGT is advisable within CG training for similar goals. However, if the focus is on pelvic stability and overall weight-bearing capacity, the ICGT within the CGT is preferable. The results suggest that, since the aspects of improvement continue to vary depending on the gait training methods, no statistically significant differences are considered to exist between groups for most of the variables. In particular, the ICGT appears numerically superior in improving gait variables and balance capacity, suggesting enhanced efficacy in supporting weight transfer, load distribution, and sensory feedback" (Discussion, page 10, line 302-312)

3-2. Currently, there is no established MCID research for the gait parameters in subacute stroke patients, excluding walking speed, which prevents us from presenting data for the other variables. Therefore, regarding the MCID for walking speed, we have revised the previous statement as follows:

"This resulted in an overall post-intervention increase in gait speed of approximately 0.4 m/s across all groups [26,27]. Tilson et al. (2010) estimated 0.16 m/s improvement in gait speed as the minimal clinically important difference for subacute stroke interventions, which supports the clinical significance of these improvements in both SGT and CGT [28]." (Discussion, page 9, line 257-261)

Comments 4. Discussion: Highlighting on the implications for physical therapists and rehab professionals would be beneficial for these professionals to possibly implement results into practice.

Response 4: Thank you for the detailed point-out. We have revised the discussion section as follows to emphasize the advantages of applying the findings clinically.

"Overall, gait parameters showed only minor differences between groups, ranging from 3% to 11%, with gait speed differences of up to 0.2 m/s, suggesting that all gait training methods led to positive improvements for patients with subacute stroke. For targeted outcomes, the SGT is recommended for enhancing knee stability, while OCGT is advisable within CG training for similar goals. However, if the focus is on pelvic stability and overall weight-bearing capacity, the ICGT within the CGT is preferable. The results suggest that, since the aspects of improvement continue to vary depending on the gait training methods, no statistically significant differences are considered to exist between groups for most of the variables. In particular, the ICGT appears numerically superior in improving gait variables and balance capacity, suggesting enhanced efficacy in supporting weight transfer, load distribution, and sensory feedback." (Discussion, page 10, line 302-312)

Comments 5. Limitations: The study acknowledges some limitations, but further elaboration on how the exclusion of upper body movements may have impacted the results (Discussion, Page 9)

Response 5: Thank you for the detailed point-out. The movement of the upper limb has a positive correlation with walking speed; however, when applying this to patients with stroke, considering upper limb movement may not significantly change the results. Therefore, we have removed the previously written limitations regarding the consideration of upper limb movement. (Discussion, page 10, line 313)

Comments 6. Minor Edits: Table 1: Consider including confidence intervals for muscle activity and gait parameters to provide a clearer picture of the variability in the results (Results, Page 6, Table 1)

Response 6: Thank you for the detailed point-out. We noticed that the comment refers to Table 1, but in the Results section, we mention Table 4. For Table 4, we initially considered presenting the data graphically to include confidence intervals, given the table’s space limitations. However, because significant differences were observed across all variables, rather than just specific ones, we decided to retain the numerical format. Maintaining this format is essential for understanding the degree of improvement across each variable pre- and post-intervention, as it supports a clearer interpretation of the results. We will consider graphical representation and the inclusion of confidence intervals in future studies.

Reviewer 2 Report

Comments and Suggestions for Authors

Review of the Paper: "Effects of Foot Position-Based Gait Training on Muscle Activity, Gait Parameters, and Balance in Subacute Stroke Patients"

The paper presents a valuable study investigating the effects of foot position-based gait training on muscle activity, gait parameters, and balance in subacute stroke patients. However, there are several areas that could be improved to enhance the overall quality of the manuscript.

1.      The introduction section is concise but lacks sufficient depth, particularly in terms of the literature gap and the novelty of the current study. The authors should expand this section to provide a more detailed background on the current state of research, clearly highlighting the gap in the existing literature that this study aims to address. Furthermore, a stronger emphasis on the novelty of the study would help clarify its significance. For example, detailing why foot position-based gait training is an innovative approach compared to traditional methods could reinforce the importance of this research.

  1. The references are relatively limited and do not adequately reflect the most recent developments in the field. To strengthen the manuscript, the authors should incorporate more recent and relevant studies, such as those from 2024. Notable references that could improve the manuscript should include:
    1. Khaliliyan H, Bahramizadeh M, Kashani RV, Vahedi M. Effects of custom mold with peripheral textured surface foot orthosis on balance and physical function in subjects with chronic ankle instability. Adv Med Psychol Public Health. 2024;1(2):74-81. Doi: 10.5281/zenodo.10637441.
    2. Pecold J, Pruc M, Nucera G, Kurek K, Szarpak L, Al-Jeabory M. Intra-articular versus intravenous tranexamic acid in total hip arthroplasty: A systematic review and meta-analysis of randomized controlled trials. Adv Med Psycol Public Health. 2024;1(4):185-198. Doi:10.5281/zenodo.11075371.

3.      While the paper touches upon important aspects of rehabilitation in subacute stroke patients, it could benefit from a clearer articulation of its contribution to the field. The authors should explain how their approach adds value to current rehabilitation practices. For instance, comparing the outcomes of foot position-based gait training to conventional methods or other innovative gait training interventions would strengthen the argument for the study’s relevance.

Overall, this study provides meaningful insights into the potential benefits of foot position-based gait training for subacute stroke patients.

However, the paper would benefit from a more comprehensive introduction, the inclusion of updated references, and a clearer explanation of the study’s novelty and clinical relevance.

In my viewpoint strengthening these aspects will make the manuscript more robust and impactful in the field of stroke rehabilitation.

Author Response

Comments 1. The introduction section is concise but lacks sufficient depth, particularly in terms of the literature gap and the novelty of the current study. The authors should expand this section to provide a more detailed background on the current state of research, clearly highlighting the gap in the existing literature that this study aims to address. Furthermore, a stronger emphasis on the novelty of the study would help clarify its significance. For example, detailing why foot position-based gait training is an innovative approach compared to traditional methods could reinforce the importance of this research.

Response 1: Thank you for the detailed point-out. This study addresses the gap in previous research, which has primarily focused on analyzing gait without considering clinical interventions. Given the lack of direct interventions and comparative analyses, our research aims to conduct interventions and compare the differences before and after, specifically examining the distinctions between straight walking and curved walking, as well as the effects of foot placement on curved walking. Therefore, the following text has been added and revised (Introduction, Page 2, line 61-72) :

“Lee (2024) demonstrated that, compared to healthy controls, patients with stroke require phase-specific rehabilitation strategies during CG, due to alterations in spatial synergy during the swing phase and temporal synergy during the swing-stance transition phase, depending on the walking direction [12]. The motor challenges encountered during CG in patients with stroke underscore the necessity for targeted training protocols tailored to CG patterns. Previous studies indicate that gait mechanics and functional adaptations vary significantly with foot placement and directionality in CG, highlighting the need to consider these variables in rehabilitation planning. However, recent studies have primarily focused on analysis without empirical investigations assessing the clinical application of interventions in patient populations. The objective of this research is to evaluate the effects of training based on the positioning of the affected foot, comparing the impact on gait in both the CG and SG.”

Comments 2. The references are relatively limited and do not adequately reflect the most recent developments in the field. To strengthen the manuscript, the authors should incorporate more recent and relevant studies, such as those from 2024. Notable references that could improve the manuscript should include

Response 2: Thank you for the detailed point-out. The existing research on curved walking has been previously studied, and the papers referenced in that research are the most relevant. A review of the most recent studies shows that, aside from a few additional citations (Introduction, page 2, line 61/ Discussion, page 9, line 280), most do not provide interpretations or necessary insights related to the findings of this study.

Comments 3. While the paper touches upon important aspects of rehabilitation in subacute stroke patients, it could benefit from a clearer articulation of its contribution to the field. The authors should explain how their approach adds value to current rehabilitation practices. For instance, comparing the outcomes of foot position-based gait training to conventional methods or other innovative gait training interventions would strengthen the argument for the study’s relevance

Response 3: Thank you for the detailed point-out. We have revised the discussion section to emphasize the advantages of applying the findings clinically. Additionally, since the improvements in muscle activation and gait variables ultimately lead to enhancements in dynamic balance ability, specifically measured by the Timed Up and Go (TUG) test, we have added a comparison between the latest research on TUG and the results of this study.

  • "However, the SGT proved more effective in enhancing the muscles that contribute to knee stability, with VL and VM muscle activity increasing by 25.5% and 5.1%, respectively, indicating a particular advantage of SG in this area. Within the CG group, the OCGT showed greater improvements in VL and VM muscle activity, with increases of 2.6% and 23.6%, respectively, compared to the ICGT, suggesting that the OCGT places higher stability demands on the outer leg." (Discussion, page 9, line 261-266)
  • "In summary, these results suggest that SG is optimal for enhancing knee stability, while CG training, particularly ICGT, provides greater benefits for improving weight-bearing capacity. Therefore, ICGT is recommended when the goal is to enhance weight transfer, load distribution, and sensory feedback, which may further contribute to improvements in balance and overall gait performance." (Discussion, page 9, line 273-277)
  • "In the TUG test, which assessed balance ability, all groups demonstrated significant reductions in test times post-intervention, indicating improvements in dynamic balance. Pournajaf et al. (2023) and Lin et al. (2022) reported improvements in TUG ranging from 12.6 to 13.4 seconds following 15 to 20 sessions of robot-assisted gait training. Similarly, in this study, the ICGT exhibited a reduction of 12.5 seconds, reflecting effects comparable to robot-assisted training [29,30]." (Discussion, page 9, line 278-283)
  • "Overall, gait parameters showed only minor differences between groups, ranging from 3% to 11%, with gait speed differences of up to 0.2 m/s, suggesting that all gait training methods led to positive improvements for patients with subacute stroke. For targeted outcomes, the SGT is recommended for enhancing knee stability, while OCGT is advisable within CG training for similar goals. However, if the focus is on pelvic stability and overall weight-bearing capacity, the ICGT within the CG training is preferable. The results suggest that, since the aspects of improvement continue to vary depending on the gait training methods, no statistically significant differences are considered to exist between groups for most of the variables. In particular, the ICGT appears numerically superior in improving gait variables and balance capacity, suggesting enhanced efficacy in supporting weight transfer, load distribution, and sensory feedback." (Discussion, page 10, line 302-312)